# The Kirkhouse Trust: Successes and Challenges in Twenty Years of Supporting Independent, Contemporary Grain Legume Breeding Projects in India and African Countries

**DOI:** 10.3390/plants13131818

**Published:** 2024-07-01

**Authors:** Claudia Canales Holzeis, Paul Gepts, Robert Koebner, Prem Narain Mathur, Sonia Morgan, María Muñoz-Amatriaín, Travis A. Parker, Edwin M. Southern, Michael P. Timko

**Affiliations:** 1The Kirkhouse Trust, Unit 6 Fenlock Court, Long Hanborough OX29 8LN, UK; mockbeggars@gmail.com (R.K.); mathur.prem@outlook.com (P.N.M.); sonia.morgan@kirkhousetrust.org (S.M.); ed.southern@bioch.ox.ac.uk (E.M.S.); 2Section of Crop & Ecosystem Sciences, Department of Plant Sciences, University of California, 1 Shields Avenue, Davis, CA 95616, USA; plgepts@ucdavis.edu (P.G.); trparker@ucdavis.edu (T.A.P.); 3Departamento de Biología Molecular (Área Genética), Universidad de León, 24071 León, Spain; mmuna@unileon.es; 4Department of Biology, University of Virginia, Charlottesville, VA 22904, USA; mpt9g@virginia.edu

**Keywords:** legumes, crop breeding, Africa, cowpea, common bean, dolichos lablab, Bambara groundnut, marker assisted selection

## Abstract

This manuscript reviews two decades of projects funded by the Kirkhouse Trust (KT), a charity registered in the UK. KT was established to improve the productivity of legume crops important in African countries and in India. KT’s requirements for support are: (1) the research must be conducted by national scientists in their home institution, either a publicly funded agricultural research institute or a university; (2) the projects need to include a molecular biology component, which to date has mostly comprised the use of molecular markers for the selection of one or more target traits in a crop improvement programme; (3) the projects funded are included in consortia, to foster the creation of scientific communities and the sharing of knowledge and breeding resources. This account relates to the key achievements and challenges, reflects on the lessons learned and outlines future research priorities.

## 1. The Origin of the Kirkhouse Trust

The Kirkhouse Trust (KT) is a UK-registered charity founded by Sir Ed Southern to fund the improvement of legume crops that are important for food and nutrition security in African countries and India and to promote scientific education. The origins of KT are entwined with the development of Sir Ed’s molecular biology company, Oxford Gene Technology (OGT). In 1997, Oxford University assigned Sir Ed’s microarray patents to OGT in exchange for 10% of the equity. In 2000, OGT’s income began to grow, and KT was registered as a charity and endowed with an initial donation from the company. 

Sir Ed Southern was persuaded to focus on plant sciences because this field is typically neglected both by governments and the UK charity sector. KT was to concentrate on crop improvement in developing countries with respect to a small number of crops and spend its resources proactively over a period of ~15 years. KT also promoted the use of molecular biology tools to accelerate the development of improved crop varieties. The goals of the very first molecular marker workshop conducted by KT carried out at the University of Agricultural Sciences at Bangalore (India) in 2003 were not only to expound the principles of marker usage in crop breeding but also to establish a working molecular marker laboratory. This facility remains in existence to this day and is used both for teaching and breeding work. A notable example of its usage has been given by Dinesh and his colleagues, who have used it to breed for disease resistance in cowpea (*Vigna unguiculata*). 

The strategy for KT was designed to build a network of scientists to generate a research community. Since sustained, long-term funding is needed to ensure that any investment in facilities is not wasted due to lack of continued support, KT has aimed to maintain the essential laboratory and operational infrastructure initially provided to projects and to enlist the cooperation of the ‘host’ institute. KT has inevitably taken a risk by investing in projects sited in countries with limited resources and volatile politics. However, if scientific capacity is to be built in a region, it needs to be driven by indigenous scientists. In 2020, KT became a charity in perpetuity, spending the income of its assets to fund its operations.

Legumes important in the target countries were selected as the focus for several reasons: (1) tropical legumes can produce an acceptable harvest in very harsh conditions with relatively limited agricultural inputs, making them suitable for climate change adaptation; (2) their grains are a good source of dietary protein and essential micronutrients, and they can also be traded in local markets to provide a source of rural income; (3) legumes have the ability to establish symbiotic relationships with nitrogen-fixing soil bacteria, improving soil fertility and reducing the farmers’ dependency on synthetic inorganic fertilisers; and (4) their leftover biomass provides a good quality feed for livestock in the dry season.

## 2. KT’s Funding Model

KT’s funding model aims to address its twin objectives of improving legume crops, which are important for smallholder farming systems in target countries and raising national scientific capacity. KT has a hands-on strategy, with a team of international scientific consultants working closely with the PIs and students they mentor, providing technological backup as needed, and hosting PIs and students for study visits in their laboratories. Since the recipients of awards must be able to do all their work in their host institution, KT funds the establishment and maintenance of essential infrastructure (including laboratories, greenhouses, and screening facilities). The support provided is intended to be long-term after an initial one-year pilot project, awarding three-year rolling grants to successful projects. To promote the establishment of international scientific communities and to encourage collaboration and the sharing of tools and resources, KT-funded projects are grouped into consortia, focusing on the same crop and similar or complementary problems. Once a year, KT convenes an in-person meeting (online during the pandemic years) for KT grantees to share progress and interact with each other.

This article reviews the successes of the first two decades of KT-funded projects (Figure 1). The individual sections focus on the achievements of the early breeding projects funded in India on dolichos lablab (*Lablab purpureus*), the cowpea (*Vigna unguiculata*) and common bean (*Phaseolus vulgaris*) breeding consortia in Africa, and a collaborative Indo-African initiative set of projects aiming to establish the potential of a selected group of orphan legumes for climate change adaptation. The paper concludes with some reflections regarding the main challenges encountered and the lessons learned.

## 3. The Dolichos Lablab Breeding Programme at the University of Agricultural Sciences, Bangalore, India

The University of Agricultural Sciences in Bangalore (UAS-B) was the first recipient of KT’s funding, which supported a series of research projects between 2002 and 2019, focusing on an array of legume crops, along with assorted infrastructure projects. The dolichos lablab breeding programme led by M. Byre Gowda has had a major impact on farmer livelihoods in South India, thanks largely to his recognition that converting what was a photoperiod-sensitive crop into an insensitive one would deliver flexibility with respect to the crop’s planting time, to the extent that more than one crop could be grown in a given year. Until the release of the HA4 and HA5 photoinsensitive varieties bred by the UAS-B programme ([1,2]; Figure 2), local farmers grew a single crop per year of photoperiod-sensitive, indeterminate types. Indeterminacy has traditionally been favoured by subsistence farmers as it enables pods to be harvested over a lengthy period, but the increasingly market-led economy of India requires year-round production, which can only be achieved using photoperiod-insensitive cultivars. The HA varieties, in addition to their photosensitivity, produced determinate plants, thereby offering an opportunity to switch from manual to mechanical harvesting.

Alongside the breeding programme, KT also supported the establishment of a germplasm collection at UAS-B, which by 2016, had grown to include about 650 accessions, representing the largest ex-situ collection of this species in the world [3]. A substantial effort has been dedicated to characterising and evaluating the accessions with respect to a wide range of vegetative, inflorescence, pod and seed traits [1]. A core set of 64 has since been assembled [4] to simplify the analysis of the species diversity. Between 2008 and 2011, KT also funded a project on dolichos lablab at Moi University, Kenya, under the direction of Miriam G. Kinyua [5,6,7,8], which resulted in the release of four varieties in 2015: Eldo KT Black 1, Eldo KT Black 2, Eldo KT Cream and Eldo KT Maridadi (https://www.uoeld.ac.ke/research/wheat-and-dolichos-varieties-research-university-eldoret; accessed on 26 June 2014) [9].

## 4. The West African Cowpea Consortium (WACC): From Conception to Improved Crops in the Field

### 4.1. Origins of the WACC

Cowpea is a widely cultivated legume species in over 65 countries, covering Africa, Asia, the Middle East, Southern Europe, and the Americas [10]. It has a high content of protein in its seeds (25% to 30% by weight), mineral-rich leaves, and abundant straw, making it a major source of food for humans as well as livestock in resource-challenged environments. Because of its ability to fix atmospheric nitrogen, it requires less fertiliser and can contribute to soil fertility. Moreover, it generally withstands high temperatures and low water availability, making it a climate-resilient crop in drought-prone areas [11]. Today, it is estimated that cowpea is cultivated on over 10.6 Mha worldwide, with an annual production of over 7.4 Mt, of which nearly 5.2 Mt is produced in Africa. Cowpea is especially important in sub-Saharan West Africa, which accounts for over 80% of its production, although with climate change constraining legume production across the continent, increased production of cowpea can now be seen in other regions, especially East and Southern Africa. Although cowpea is an African crop in origin, it has been introduced across the globe [12]. It is assumed to have been introduced to Brazil in the 16th century, where its cultivation has expanded rapidly. Cowpea is now the second most consumed type of bean in Brazil, and it is being cultivated in the North, Northeast, and Centre-West regions. Brazil is now the second most important producer of cowpea grain.

Historically, the average yield of cowpeas was relatively low, with traditional landraces and cultivated forms yielding at most 400–500 kg/ha. This was due to a lack of genetic improvement since domestication and the large number of diseases, pests and parasites that constrained its growth and productivity. Although genetic improvement programmes began with the Green Revolution in the late 1960s, little progress was realised until cowpea was listed among the mandated target crops for improvement by the International Institute of Tropical Agriculture (IITA). Under the direction of Professor B.B. Singh and others, the IITA was able to collect and evaluate cowpea germplasm from across Africa and begin systematically breeding for disease and pest resistance as well as farmer-desired agronomic traits such as larger seed size, photoperiod insensitivity and rapid maturity. Breeding activities focused on stacking these various farmer-preferred traits, which using conventional selection and breeding techniques available at the time, was a long and tedious process taking years to decades to realise the release of improved varieties.

With the advent of molecular genetics in agriculture, emphasis was placed on leveraging genomic information for rapid improvement in crops through molecular-based breeding. Initial applications of marker-assisted selection and breeding platforms were targeted at high-value crops like maize, rice, wheat, and soybean. Consequently, few resources were allocated to developing the genomic scale resources necessary for the improvement of cowpea, a crop thought to have little socioeconomic importance outside of West Africa. With a greater appreciation of cowpea’s value as a climate-resilient grain legume and the recognition of the broader importance of the crop to the lives and livelihoods of small shareholder farmers in resource-limited Sub-Saharan Africa, from the early 2000’s onward, an intensive effort was undertaken to improve genomic resources for cowpea leading to greater ability to speed the delivery of improved cultivars for use in low input agricultural production in Africa. This switch coincided with decreased costs for genomic scale sequencing, thereby allowing for the development of the molecular tools needed.

Among the first attempts at characterising the gene content and complexity of the cowpea genome was a KT-supported project initiated in early 2006 with the University of Virginia targeting the hypomethylated, gene-rich coding sequences in the genome of the African cowpea cultivar IT97K-499-35 via the use of methylation filtration (MF), a reduced representational sequencing approach known to overcome the presence of ubiquitous repetitive DNA. Using MF, Timko et al. [13] were able to achieve a 4.1-fold enrichment for the gene-rich space of cowpea and generated 263,425 gene-space sequence reads (GSRs) that could be assembled into 41,260 unigenes representing 19,786 unique GenBank accession numbers [14]. The presence of >1000 simple sequence repeats (SSRs) in the genic regions was exploited in a KT-funded programme dubbed the West African Cowpea Consortium (WACC), which sought to integrate comparative genomics for molecular marker discovery, marker validation through coordinated participatory field testing of markers by members of the consortium, and capacity building through training in molecular biology, breeding, and bioinformatics. At its establishment in March 2007, the WACC included research partners from Nigeria, Burkina Faso, and Cameroon, later expanded to include breeders from Mali, Niger, Senegal, Ghana, Togo, and Benin.

At its core, the WACC was dedicated to the development of tools and techniques for marker-assisted breeding (MAB) that are affordable, readily available and easily applicable in African laboratory settings. Rather than tackle the many biotic and abiotic constraints that limited cowpea productivity, a conscious decision was made to first focus on a small set of major constraints under simple genetic control to accelerate the improvement of local germplasm and train local breeders.

### 4.2. Achievements of the WACC

Among the early successes of the WACC was the capacity to improve local varieties for resistance to *Striga gesnerioides*, a parasitic plant which attaches itself to the cowpea root and severely constrains yields throughout the Sahel. Studies carried out in the early 2000s identified amplified fragment length polymorphism (AFLP) markers associated with genes conferring resistance to *S. gesnerioides* races SG1 and SG3 [15,16,17,18]. These AFLP markers were used to generate the sequence-characterised amplified region (SCAR) markers 61R and 61RM2 (MahSE2) linked to a gene(s) conferring resistance both the SG1 and SG3 as [17], allowing for the convenient selection of resistant cultivars [10].

The development of a substantial panel of SSRs (see above) and the identification of SSR-1, a sequence embedded in the gene conferring resistance to SG3, the dominant race of *S. gesnerioides* found in West Africa [19], proved pivotal in the implementation of broad-scale MAB. The ability to use SSR-1 in molecular breeding-based screening of germplasm using simple PCR and gel electrophoretic technologies allowed researchers within the WACC to rapidly improve *Striga*-susceptible local farmer-preferred cultivar for resistance. Subsequent studies identifying germplasm resistant to the various distinct races of *S. gesnerioides* present across West Africa and the development of SSR molecular markers able to select for these resistances [20,21] resulted in the development of *Striga*-resistant cultivars useful throughout the WACC footprint. The use of SSR-1 (race SG3) and the two SCAR markers C42B (SG 5) and 61R/Mahse2 (SG1 and SG2) have been used for breeding for *Striga* resistance in Burkina Faso, Nigeria, Cameroon, Mali and Ghana. Among the successful releases are the cultivars FUAMPEA 1 to FUAMPEA 4 (developed by Lucky Omoigui and his team at UAM in Nigeria [22,23]), ‘Zaayura Pali’, ’Soo-Sima’, ’Diffeele’, ’Wang Kae’ and ‘Kirkhouse Benga [22]’ developed by Francis Kusi and his colleagues at Savanna Agricultural Research Institute (SARI) in Ghana, Komcallé, Tiligré, Nafi, and Gourgou developed by Benoit Batieno and his colleagues at INERA [24], Burkina Faso, IR15-MA-02 and IR15-MA33 developed by Sobda Gonné at IRAD, Cameroon and CZ06-3-1 (Acar 1), CZ06-2-17 (Simbo), CZ06-1-12, CZ06-4-16, and CZ06-1-05 developed by Mamadou Touré, Sory Diallo and their coworkers at IER, Mali [22].

Since these initial studies, the continued development and use of molecular marker technologies for molecular breeding activities within the WACC permitted an expansion of the types and number of constraints that could be selected for improvement. This led to the identification of new resistance-linked markers and the breeding of new cowpea varieties with stacked resistances to *Striga* [25,26], *Alectra vogelii* [27], aphids (*Aphis craccivora*) [28], brown blotch (*Colletotrichum capsici*) [29] and *Fusarium* [30] (Table 1).

Coincident with this was the application of emerging platforms for high throughput DNA and cDNA sequencing and single nucleotide polymorphism (SNP) detection in cowpea by Muchero et al. [31]. These researchers were able to map 928 expressed sequence tag (EST)-derived SNPs using an Illumina 1536 GoldenGate platform. Building upon this work, Lucas et al. [32] developed a new consensus map containing 1107 EST-derived SNP markers developed by integrating 13 population-specific maps. The use of SNP technologies changed how molecular breeding in cowpeas could be accomplished and demonstrated the effectiveness of integrating SSR and SNP markers for trait mapping and marker-assisted breeding.

With the emergence of resistance to aphids as a priority among WACC partners Fransis Kusi and his colleagues at the SARI, Ghana, began studies aimed at the development of robust screening techniques useful in the identification of new sources of resistance. Using improved screen house and field phenotyping, Kusi and his group [33] identified SARC 1-57-2 as a cowpea breeding line containing a novel aphid resistance locus. Using a panel of SSRs developed within the WACC, they were able to identify a co-dominant SSR marker (CP 171F/172R) that segregated with aphid resistance with a recombination fraction of 5.91% in an F2 population derived from crossing Apagbaala x SARC 1-57-2. Using CP 171F/172R for foreground selection and a KASP-SNP marker panel for background selection in marker-assisted backcrossing, the resistance from SARC 1-57-2 was introduced into elite susceptible cultivar Zaayura [33].

Subsequent genomic scale sequence characterisation and transcriptomic profiling resulted in the development of high-quality genetic maps and genome sequence assemblies that permitted the development of effective molecular markers for a broader set of biotic constraints within the WACC. A genetic linkage map based on the segregation of simple sequence repeat (SSR) markers developed using a recombinant inbred (RI) population derived from a cross between the California Blackeye type 524B and 219-01, perennial wild cowpea from Kenya allowed Andargie et al. [34] to map agronomic traits related to domestication such as seed weight and pod shattering, as well as floral scent characteristics.

With the advent of new and improved technologies came the development of platforms for high throughput SNP detection, beginning with the Illumina 1536 GoldenGate platform developed by Muchero et al. [31] with 928 expressed sequence tag (EST)-derived SNPs and culminating with the 51,128-SNP Cowpea iSelect Consortium Array [35]. This time frame also saw the refinement of resources for accurate genome annotation, identification of candidate genes, and comparative genomic analysis, pushing cowpea functional genomics to new levels of utility [36,37,38,39]. Using these improved SNP data, Dr. Erik Ohlson at UVa created a panel of ~200 well-spaced allele-specific SNPs, aiming to provide the WACC laboratories with a quick way of locating the approximate chromosomal location of resistance genes using PCR and gel electrophoresis. One could then identify additional SNPs in that region to identify more tightly linked markers as needed. Most recently, a mid-density marker platform for cowpea genotyping has been developed that contains 2602 SNP markers with an average density of about 4 SNPs/Mbp throughout the 11 cowpea chromosomes [40]. This genotyping platform, based on diversity array technology (DArT), is currently the most useful low-cost, high throughput method for rapid identification of new markers for key constraints.

In 2023, the WACC expanded its geographic focus, engaging researchers in Malawi, Zambia, Zimbabwe, and other locations in Southeastern Africa, and was renamed as the African Cowpea Programme (ACP). This expansion was conducted with the hope of transferring molecular tools, breeding materials, and technical know-how from the WACC to speed the development of improved cultivars for regions across the African continent.

### 4.3. Future Priorities for the African Cowpea Programme (ACP; Formerly WACC)

The ACP will continue to support the deployment of marker-trait associations in the selection of breeding lines, speeding up the development of improved germplasm for local markets. A priority will be to facilitate the transmission of knowledge between programmes, with markers and germplasm already developed by prior WACC projects being transferred to newly funded teams in Southeastern Africa. This includes markers and donor lines carrying resistance alleles to several biotic stressors such as aphids, *Fusarium* and *Striga*. The newly developed genomic resources, including the cowpea reference genome and pan-genome [39,41], will be useful in developing new markers in cases where the linkage between the available marker and the causative locus is broken. At the same time, the ACP will support the identification of new sources of resistance for similar or new biotic threats, facilitating the discovery of novel marker trait associations whenever needed.

An expansion in the number of traits targeted by the ACP is among KT’s objectives. We are aware of the importance of breeding for consumer-related traits in order to develop varieties that are accepted by the public. The genetics of many traits related to seed coat colour and patterning are better understood now (e.g., [42,43]), and the findings can be used by cowpea breeding programmes to select lines with locally preferred seed coat characteristics at an early point in the breeding cycle. The feasibility of using marker-assisted selection to increase seed size has been demonstrated by the success of Lo [44] in introgressing multiple loci into a popular Senegalese cultivar. Seed coat texture, an important quality trait influencing the end use of the seed, is expected to be under relatively simple genetic control [45]. Seed coat roughness is known to be negatively correlated with cooking time, a trait of major importance in sub-Saharan Africa, not just because firewood, charcoal and water for cooking are in short supply but also because of the importance of reducing exposure to smoke. Investments that enable the identification and deployment of markers tightly linked to loci-controlling consumer traits will help ensure that improved varieties are adopted by farmers and consumers.

Abiotic stresses, especially heat and drought, are major threats to cowpea productivity, and the frequency of their occurrence is rising as a result of climate change. Even though cowpea is well adapted to a semi-arid hot environment, extreme events, including heatwaves and long drought periods, are negatively impacting cowpea yields [46,47]. While improving adaptation to these abiotic stresses is urgently needed, their complex inheritance makes them more difficult to tackle using a conventional breeding approach. In this sense, KT´s strategy of promoting the use of marker-assisted selection to identify desirable segregants is the only effective breeding strategy for developing varieties with an increased level of stress tolerance. First, however, it will be necessary to improve our understanding of the genetic architecture of the stress response. To date, no major loci have been validated for traits related to these abiotic stresses, and the only strategy implemented by the cowpea programmes so far has been to target early flowering individuals that can escape droughts occurring at the late growth stage. The ACP is determined to take steps towards making marker-assisted breeding of heat and drought-tolerant cowpea varieties a reality in the near future. These steps include (1) connecting with funding agencies, institutions and researchers conducting work on abiotic stresses to foster discoveries through collaboration and (2) improving data collection related to field trials at many different locations in Africa (both agronomic and climate data from weather stations), which can be used to gain a better understanding of drought and heat-related traits and to help develop model-based predictions.

Capacity building and training will continue to be a priority for the ACP. The experience gained throughout the years has helped define our training priorities. Special emphasis will be put on concepts behind MAB (e.g., linkage disequilibrium), which are often incomplete. The use of markers in the selection process is an important component that all KT-funded programmes include, and gaining a deeper understanding of MAB technology would empower African breeders and scientists to fully utilise the method in their breeding programmes and troubleshoot any problem quickly and in an independent way. Another issue that is becoming increasingly challenging is the maintenance of the identity and purity of the breeding materials. The successful use of markers to guide breeding decisions depends heavily on maintaining the genetic purity of the seed. Therefore, the ACP will provide sustained support to increase awareness and endorse better practices during the process of seed increase, whether in a screenhouse or in the field.

The number and quality of the genetic and genomic resources available for this crop are expanding and will continue to be exploited by the ACP to maximise the impact of investment on cowpea improvement programmes. Multiple populations genotyped at a high density are available, including several biparental RIL populations [35], a multi-parent MAGIC population [48], and core (IITA Core [49]) and mini-core (UCR Minicore [50]) collections. The use of these resources, coupled with the germplasm developed by prior WACC projects, will surely broaden the achievements of the ACP.

## 5. The African Bean Consortium (ABC): Developing Improved Bean Varieties in the World’s Highest Per Capita Bean-Consuming Region

### 5.1. The Origins of the ABC

The ABC project sought to accomplish three complementary goals. First, it aimed to empower African bean breeders to develop their cultivars from the initial varietal concept to advanced lines ready to be submitted to national trials. Second, it sought to upgrade the capabilities of the breeding programmes so they could take advantage of genomic information for the common bean and apply markers to assist selection in the breeding pipeline [51]. Third, it allowed a more decentralised pursuit of plant breeding to take advantage of specific or local, rather than general, adaptation [52,53]. Accomplishing these goals required a careful choice of each project’s specific objectives. These were discussed at the first meeting of the ABC project, which took place in Nairobi in 2006. Participating at this meeting were representatives from bean breeding programmes in East Africa, namely from Kenya (Kenya Agricultural and Livestock Research Organization and the University of Nairobi), Malawi (Bunda College of Agriculture), Rwanda (Rwanda Agricultural Board), Tanzania (Sokoine University of Agriculture) and Uganda (National Agricultural Research Organization). Later additions to the consortium were breeders from the Southern Agricultural Research Institute (Hawassa, Ethiopia), the University of Zambia, the University of Embu (Kenya), and the Instituto de Investigação Agrária de Moçambique (Mozambique).

At the 2006 meeting, the attendees decided that the ABC would focus on introducing multiple disease resistances for several reasons. First, diseases are a nearly general production constraint worldwide where beans are produced. Introducing resistance to these multiple diseases would represent a significant improvement as each of the pathogens causes substantial yield losses [54,55]. Second, bean breeders and geneticists, mainly in the USA and Brazil, had already developed molecular markers tagging major resistance genes for several diseases (e.g., [56,57,58,59,60,61,62,63,64]). Third, for several diseases or the *Rhizobium* symbiont, a process of parallel geographic diversity, possibly due to bean-microbe or virus coevolution, had been observed [65,66,67,68,69] prior to the initiation of the project. Alongside the backcrossing of resistance genes into preferred cultivars, the ABC projects included the collection of diseased plant tissue and seed, the isolation of pathogen strains and the characterisation of the genetic diversity of both the host and the pathogen. These constituted research topics for KT-supported students pursuing a graduate degree (e.g., [70,71,72,73,74,75]) and would have important implications for resistance breeding, including information about pathogen and host diversity and choice of host resistance sources.

Five diseases, based on the significant losses they cause in East Africa and on the availability of DNA markers, were prioritised, namely BCMV/BCMNV (bean common mosaic virus/bean common mosaic necrosis virus), CBB (common bacterial blight caused by *Xanthomonas campestris*), anthracnose (caused by the fungus *Colletotrichum lindemuthianum*), ALS (angular leaf spot, caused by the fungus *Pseudocercospora griseola*) and *Pythium* root rot. Other constraints were considered but were ultimately not adopted in order to streamline the multi-project consortium. These constraints were rust (*Uromyces appendiculatus*), the insect herbivores bean stem maggot (*Ophiomyia* sp.), bruchids (*Acanthoscelides obtectus*) and aphids (*Aphis* sp.), drought stress, low soil fertility, and consumer traits such as cooking time and biofortification. Insufficient information about the genetic control of these traits and the absence of marker tags were the predominant reasons to defer breeding for these traits.

The markers available in 2006 at the initiation of the ABC project were mainly SCARs [76] derived from RAPD (random amplified polymorphic DNA) sequences, and were thus dominant. When in *cis* with the resistance gene, a dominant marker is useful to identify individuals carrying the resistance allele in successive backcross progenies. However, co-dominant markers are preferred since these allow for discrimination between homozygous and heterozygous carriers. A successful ABC-supported programme to replace a dominant with a co-dominant marker was the development of an insertion-deletion (InDel) marker tagging the *Phg-2* gene for resistance to ALS [77].

### 5.2. PhaseolusGenes, the “Bean Breeders’ Marker Toolbox” and the Research It Supported

The search for molecular markers was facilitated by the development of the PhaseolusGenes marker database as a collaboration between the Gepts group in the Department of Plant Sciences at UC Davis and the Bioinformatics Unit of the UC Davis Genome Center starting in 2008, with the assistance of D. Lin, J. Boveda, M. Briton, J. Fass, N. Joshi, Z. Lu, and A. Schaal [78]. PhaseolusGenes provided information on legacy markers [e.g., SCAR and sequence-tagged sites (STS) markers] and newly developed markers [SSRs and InDels], all anchored on a reference whole-genome sequence. Initially, this sequence was the soybean sequence (*Glycine max*), a phylogenetically relatively close taxon with a paleotetraploid genome [79], but later replaced by the first common bean reference sequence (version 1.0, developed in the Andean landrace G19833) [80].

The initial design of the database included three elements: (1) a database proper, with information on individual markers (including sequence, PCR primers, description of potential usefulness as markers, linkage map and genome locations, and source; (2) a genome browser, showing the genome location of several marker categories aligned initially against the soybean genome and, more recently, the Andean reference sequence for common bean (version 1.0, [80]); and (3) a comparative mapping instance (CMap) based on shared markers among several molecular genetic maps, useful to determine linkage relationships among widely different traits.

The genome browser included a broad range of genomic sequence types from various sources that had been collected over the years. In addition to individual tracks devoted to previously established markers, such as SCARs and STSs, the browser included tracks for ESTs, BAC-end sequences (BAT93 library [81], STSs of *P. vulgaris* [82], ESTs of other species (developed by the J. Craig Venter Institute: runner bean (*P. coccineus*), cowpea etc.), and a track for aligning common bean sequence reads containing predicted SSR motifs.

A survey of the scientific literature over the last nine years since the inception of the PhaseolusGenes database provides an assessment of the different uses of the database (Figure 3). As expected, based on its design [77,78], the PhaseolusGenes database was used repeatedly to identify markers linked to agronomic traits, mainly disease resistances, with some 65 studies citing the database. The markers developed included three for the diseases that are objectives of the ABC, namely ANT, ALS, and CBB.

Gonçalves-Vidigal et al. [96] identified two STS markers, CV542014450 and TGA1.1570, linked to a disease resistance cluster containing the *Co-1^4^* and *Phg-1* loci on chromosome Pv01. Dr. Celeste Gonçalves-Vidigal was a visiting scientist from Brazil in the Gepts group at that time. With the assistance of Dr. James Kami, she identified and validated these two markers in approximately two weeks. Prior to the existence of PhaseolusGenes, such identification would have taken several months on average. More recently, Murube et al. [94] integrated the genetic and physical map positions of ANT resistance genes on chromosomes Pv01 and Pv04.

The PhaseolusGenes database was also useful to identify markers of other genes for resistance to ALS. Miller et al. [127] identified several markers closely linked to the *Phg-2* locus on chromosome Pv08. Of these, the InDel g796 provided the most reliable and reproducible indel marker, located at 3 cM from *Phg-2*. It is noteworthy that g796 was unique to MEX54, the donor genotype of ALS resistance and differentiated not only the target varieties but also other resistance donors, making it a quasi-unique polymorphism among all genotypes tested. More recent research by Gil et al. [132], based on data from Lobaton et al. [127], has further narrowed down the chromosome fragment containing *Phg-2* to ~400 Kbp. 

In research, wholly or partially funded by KT, Okii et al. [125] studied the population structure and genetic diversity of the Ugandan common bean collection. They observed that in this collection, Andean and Mesoamerican genotypes were about equally represented. A study combining socio-economic surveys and SNP marker analysis by Wilkus et al. [128] found that farmers who had participated in regional participatory research were more likely to adopt new varieties and diversify their seed stocks. There were differences among farmers for their propensity to adopt Mesoamerican varieties, the group they were least familiar with, but which represented the largest change in bean diversity.

For a variety of reasons, including scientific (comparative genomics), organisational, and streamlining purposes [79,80,81], PhaseolusGenes was transferred into PhaseolusMine in the Legume Information System (LIS) [82] to maximise the utility of the information for the legume community and to make available all online comparative genomics tools available within LIS.

### 5.3. Accomplishments of the ABC

The principal accomplishment of the ABC project has been the development of improved varieties and advanced breeding lines that have reached the stage where they have been tested or will be shortly, by the respective national authorities for release as commercial varieties in their respective countries. These include the Ethiopian varieties ‘Key Wolaita’, an improved version of the small red landrace ‘Red Wolaita’ with ALS resistance sourced from Mex-54 and CBB resistance sourced from VAX-6, as well as ’Key Burre’, an improved version of the red mottled landrace Ibado, also with the Mex-54 ALS resistance and the VAX-6 CBB resistance; these two cultivars were released in Ethiopia in 2023 by the team of Yayis Rezene. In Uganda, NABE 12CR (NABE 12C improved with resistance to ANT) and NABE 14R (NABE 14 improved with resistance to root rot) were released in 2024 by the team of Stanley Nkalubo (Table 2). The cultivars Zerengeti (resistant to ANT) and Kundadila (fast cooking, high iron bioavailability) have been developed by Kelvin Kamfwa’s team and are expected to be released soon in Zambia after completing their last year of testing required for registration

These high-yielding, multiple disease-resistant varieties have progressed through the KT-funded breeding pipeline of each project from the initial crosses, the early-generation marker tests and through to late generation field testing.

### 5.4. Future Priorities for the ABC

Moving forward, the ABC will continue to prioritise two main focal activities: First, the breeding of new bean varieties for African smallholder farmers, and second, the training of new generations of African breeders to efficiently develop such novel varieties in the future. The African Bean Consortium’s strategy focuses on giving local breeders the tools needed to conduct all elements of the breeding pipeline, from parent selection, crossing, and evaluation through variety release. This approach strives to ensure that all rounds of selection are conducted in response to local demands and environmental conditions. It also empowers the local community to be able to deploy a broad range of strategies to address complex challenges in their areas. Ultimately, this investment is aimed to maximise nutritional security and livelihoods for smallholder farming communities across Africa.

Future breeding work will include a renewed focus on maximising yield, key disease resistances, and a small number of the most important consumer traits. This will be accomplished through hybridisation between existing preferred varieties, with the introduction of valued alleles (such as key disease resistances) from exotic sources through recurrent backcrossing. In general, early-generation selection will focus on traits that are highly heritable and/or inexpensive to screen, including seed type (e.g., seed colour, size, and shape) and disease resistance. Marker-assisted selection will focus primarily on early generations and allele introgression through recurrent backcrossing. Selection in later generations will focus on traits with lower heritability and/or higher phenotyping costs, such as yield. In all cases, breeding priorities will continue to be set by African breeders based on their local demands, with input from international consultants on methods and planning. Education of ABC teams will continue to take several forms, including training visits to international labs (e.g., UC Davis), training sessions at annual meetings for projects, Zoom training, and cross-training between teams, particularly following visits to international labs. Educational objectives include training with the latest genotyping methods, phenotyping methods, and data analysis software tools.

Several new technological opportunities for growth exist for ABC projects. Advances in genotyping methods offer the chance for funded projects to use marker types with improved linkage to variation of interest, co-dominance for identifying heterozygotes, and higher-throughput screening. Historically, ABC teams have used dominant SCAR markers, which face limitations that are overcome by newer technology. A variety of co-dominant marker types exist and have begun to be explored by programmes; these include simple InDel length polymorphisms, cleaved amplified polymorphic sequence (CAPS) markers, and allele-specific competitive-extension PCR-based assays (e.g., KASP, PACE). These latter marker types are derived from SNPs, which have become the predominant marker type used in common bean genetic studies [87], thus offering great flexibility in design and deployment [133]. They can be designed to be in extremely tight linkage with causal genetic variation and can be rapidly developed for virtually any trait of interest that has been genetically mapped in a diverse population [134]. Importantly, these assays are also co-dominant, which greatly facilitates the stabilisation of resistances, especially when multiple genes are pyramided simultaneously. KASP and PACE also have extremely high throughput, with data managed electronically and bypassing gel electrophoresis altogether, enabling the screening of hundreds of samples at a time, depending on the equipment used. To keep up with the leading advances in common bean genetics and breeding, it will be important for ABC-funded teams to have the capacity to screen markers of a variety of important and growing modern marker classes. Adopting new technologies will encounter challenges such as the shipment of specialised equipment, maintenance, and training project teams in new techniques. Overseas marker screening is also being explored. The application of optimal marker technologies by ABC research teams across Africa is a priority for the consortium.

Further opportunities exist for ABC teams related to materials sharing. The development and release of promising breeding materials by ABC programmes offer further opportunities to expand collaborations between funded projects and between countries. Certain market classes are broadly valued by populations of several African countries (e.g., Kablanketi types in Malawi, Tanzania, Zambia, and Kenya: seeds with dense purple stippling (https://www.kirkhousetrust.org/_files/ugd/b134c0_b89695e60c0b46bfb62d33c207ad1d6f.pdf, accessed on 26 May 2024); yellow beans [135]; Kalima types; sugar beans; etc.). These regions also often value similar consumer traits and suffer from similar limitations in pathogens, pests, and abiotic stress. The continued cross-sharing of materials between neighbouring countries, both as finished cultivar releases and parents for future breeding, will be another promising area for future development.

ABC projects continue to make considerable breeding progress across numerous bean market classes, traits of interest, and countries. Continued investment in the ABC programmes will include a significant training component, ensuring that funded projects have access to the most effective genotyping, phenotyping, and data analysis methods for their respective goals. Meanwhile, as new breeding materials continue to be released, the consortium will seek to maximise its impact through the sharing of materials, through cross-release into neighbouring countries where applicable and appropriate, as well as the sharing of improved lines for use as parents in continued breeding. Through this collaborative investment, the ABC seeks to continually improve the nutritional security and quality of life for smallholder farmers and their communities across the African continent.

## 6. The Stress Tolerant Orphan Legumes (STOL) Consortium

### 6.1. The Origins of the STOL Consortium

Roughly 2.5 billion people (30% of the world’s population) live in semi-arid regions, and approximately a third of these people depend on agriculture for their food security and livelihood. Crop production in these regions has always faced challenges associated with excess heat, drought, a highly variable climate, land degradation, and a loss of biodiversity, which has been exacerbated in recent times by climate change, limited access to technology, poor market linkages, weak institutions, and lack of national and international partnerships. A possible strategy to cope with climate change is to switch from the cultivation of current crops to ones which are more drought-hardy [136,137]. These include a number of minor legume crops, commonly known as orphan legumes, currently being grown to a limited extent in the drier regions of both Africa and Asia to provide a measure of food and nutritional security to households, as well as some income to farmers. These species have remained relatively neglected by both researchers and industry because of their limited economic importance in the global market.

The STOL consortium was established in 2018 under the Promoting India-Africa Framework for Strategic Cooperation Initiative in partnership with the Indian Council of Agricultural Research (ICAR). The STOL programme aims to facilitate the introduction and exchange of stress-tolerant orphan legume varieties among partnering Indian and African institutions and assess the relative response of selected species to the higher levels of abiotic stresses expected because of climate change. The species chosen were moth bean (*V. aconitifolia*), mung bean (*V. radiata*), horsegram (*Macrotyloma uniflorum*), dolichos lablab, bambara groundnut (*V. subterranean*), marama bean (*Tylosema esculentum*), tepary bean (*P. acutifolius*), rice bean (*V. umbellata*), pigeon pea (*Cajanus cajan*), lima bean (*P. lunatus*) and adzuki bean (*V. angularis*), with cowpea included as the reference crop. The selection of the varieties and germplasm of the crops identified from India was based on their performance in the drier parts of the country, especially in western Rajasthan, where the climatic conditions closely match those of the target sites in Africa (in the Sahel; Figure 4). It was agreed that following an initial trial of two consecutive years, only those sites where the performance of the STOL crops was promising (compared to cowpea) would be selected for the subsequent testing and identification of promising varieties to be submitted for release for wider cultivation.

To resolve the institutional difficulty associated with seed exchange between India and African countries and vice versa, the project took advantage of a recently established India-Africa Strategic Partnership, a multi-dimensional South-South cooperation arrangement. Under this agreement, the Indian Government and the African Union signed a memorandum of understanding (MoU) agreeing to increase cooperation to improve the productivity, nutritious quality, and resilience of local and traditional food systems while preserving biodiversity. In 2018, an MoU was also concluded between KT and the Indian Council of Agricultural Research-National Bureau of Plant Genetic Resources (ICAR-NBPGR). Under this agreement, nine African countries (Burkina Faso, Ghana, Kenya, Mali, Namibia, Niger, Nigeria, Tanzania and Uganda) and India agreed to collaborate in a number of activities, namely to: (i) identify a panel of fifty accessions of each STOL species; (ii) multiply and distribute seed to the various project partners for evaluation; (iii) carry out field trials of the best-performing species and accessions to submit proposals for release of varieties; and (iv) train extension workers and farmers in the management of these crops. Senegal joined the STOL consortium in 2020.

### 6.2. STOL Achievements

A collection comprising 1879 accessions (Figure 5) was assembled through correspondence and the signing of a material transfer agreement (MTA) with the Nelson Mandela African Institute of Science and Technology (NM-AIST, Tanzania). Selected accessions were planted at NM-AIST, Moshi, for field evaluation during the 2017–18 growing season, and early and late maturing accessions were selected for subsequent studies.

Under the KT-ICAR agreement, systematic selection and exchange of promising germplasm from India were identified and shared under the agreed MTA (Figure 6).

Following this exchange, germplasm was planted for field evaluation using an augmented block design, applying standard practices and including four local cowpea varieties as a check to compare the performance of the new crops across all locations. The field evaluation was continued for two consecutive years (2018–2019 and 2019–2020). During each cycle of field evaluation, the best-performing accessions were selected, and poorly performing accessions were dropped, as shown in Figure 7.

A participatory selection approach was followed for the selection of promising varieties by organising farmers’ field days (Figure 8). Most of the farmers who participated in field days showed interest in trying these new crops, either for grain purposes or as fodder crops. Mung bean was considered the best grain crop by all farmers across all countries, followed by moth bean in selected sites where the average yearly rainfall lay below 500 mm. Horsegram and dolichos lablab were identified as good sources of fodder. Based on the field evaluation, 4–10 of the best-performing varieties of each crop were identified to be tested at farmers’ fields and research stations across diverse agroclimatic zones to make recommendations for the release of varieties following national variety release guidelines.

Based on the sustained performance over the years and across sites, two countries (Burkina Faso and Senegal) were considered most suitable for the cultivation of the STOL crops among the ones initially selected. Further support was provided, aiming to formally release varieties of mungbean, moth bean, and dolichos lablab at the national and provincial levels. The other former STOL country partners are conserving the germplasm introduced from India in their respective genebanks and will look for opportunities for their use in national crop improvement programmes as and when necessary. Promising germplasm was identified in Burkina Faso based on agro-morphological traits and genotype x environment interactions data collected from five locations (Kamboinse, Mani, Saria, Tita and Ziniare). In general, all varieties’ performances varied from one site to another, indicating an important role of the environment in the performance of the selected mungbean varieties (Figure 9). AVMU1621 outperformed all other varieties based on the average performance across all five locations, while Ganga-1 and AVMU1621 were the top performers in the driest part of the country (Mani). Participatory varietal selection identified two varieties of dolichos lablab (IC-0623093, HA-3) and three varieties of mung bean Ganga-1, IC-103245 (received from NBPGR) and AVMU1621 (received from the World Vegetable Center) as suitable for formal registration and their release is expected in 2025. Further trials are underway for selected moth bean accessions, with the aim of releasing suitable varieties in the coming years. In Senegal, four NBPGR mungbean accessions (Ganga-8, GAM-5, MH-421 and IC-39383) and three NBPGR moth bean accessions (RMO-4-1-6-9, RMO-25 and GMO-2) were identified as promising varieties based on their performance across five locations and are considered as suitable for formal registration and their release in 2025.

In Senegal, ten accessions of mungbean (IC-39383, IC-39375, IC-39352, MH-421, MH-14, IPM2-14, GAM-5, Ganga-8, Berken and Mam) and nine of moth bean (GMO-2, Maru Moth, RMB-28, RMO-257, RMO-435, RMO-225, RMO-25, RMO-3-5-70, and RMO-4-1-6-9) were tested across five locations viz., Bambey Research Station, three villages in Louga District (Dara Ndiakhour1, Dara Ndiakhour2, Dara Ndiakhour3) and one village in Mekhé District. All sites fall in dryland zones in Senegal with water scarcity due to low rainfall (less than 500 mm/year) and frequent occurrence of drought spells (Figure 10).

Data collected for various quantitative traits were subjected to an analysis of variance, followed by a pooled analysis of variance using a linear mixed model. A two-dimensional GGE biplot was generated using the first two principal components. To assess the effect of genotype by Environmental interaction (GEI) and evaluate the stability of productivity of the mungbean and moth bean genotypes, an AMMI model (Additive Main effects and Multiplicative Interaction) was used. Using graphical representation, the best genotypes of each specific environment were selected with broad adaptability (mega-environment) or narrow adaptability (Figure 11).

Four mungbean accessions (Ganga-8, GAM-5, MH-421 and IC-39383) and three moth bean accessions (RMO-4-1-6-9, RMO-25 and GMO-2) were identified as promising varieties based on their performance across five locations and are considered as suitable for formal registration and their release in 2025.

Marama bean is a Southern African drought-tolerant legume that produces both seed and tubers. In addition, since a plant produces numerous prostrate stems that grow up to 3 m in length, its cultivation in arid areas may contribute to preserving soil structure. A set of 521 accessions was collected from 38 distinct target sites in Namibia. Annual rainfall across collecting sites varies from 224 mm to 458 mm. However, most of the collecting sites had an average rainfall ranging between 350 to 420 mm (Figure 12).

The accessions were planted in the field for maintenance and characterisation. A set of 50 promising accessions were identified based on their performance with respect to various agro-morphological traits. Significant variation was recorded for seed yield and 1000 seed weight (Table 3). Seed yield per plant varied from 130 g to 660 g (Figure 13). All these accessions are being maintained in the field, and seed samples are also being stored at the Namibia University of Science and Technology. Of these, 18 promising accessions have been identified as either high (MBPCC225, MBPCC226, MBPCC246, MBPCC248, MBPCC382, MBPCC423, MBPCC424, MBPCC488, MBPCC501) or low seed (MBPCC1, MBPCC3, MBPCC6, MBPCC114, MBPCC161, MBPCC201, MBPCC300, MBPCC401, MBPCC405) seed producers and shared with STOL partners, with trials in India currently underway.

In addition to the above crops, a programme of introduction and testing for the adaptability of tepary bean (germplasm provided in 2017 by the Phaseolus gene bank at the Centro Internacional de Agricultura Tropical (CIAT) in Cali, Colombia) was initiated in both India and Burkina Faso [138], in partnership with University of California, Davis, USA. Under this programme, 26 selected tepary germplasm and improved lines were shared with the STOL partners and are currently under field evaluation.

STOL researchers compiled monographs commissioned by KT, focusing on dolichos lablab, horsegram, marama bean, moth bean and tepary bean, which are in the final stage of publication.

## 7. The Bambara Breeding Initiative (BBI)

One of the intended outcomes of the STOL consortium was to identify a crop to become the focus of breeding projects funded by KT, and the choice fell on Bambara groundnut, an African species which has been cultivated across the continent for centuries in low-input agricultural settings; the crop is currently also being grown in Southeast Asia.

The plant is adapted to a wide range of agroecological conditions, and since it is both highly tolerant of drought and high temperatures and its grain is very nutritious, the species provides very good opportunities for climate change adaptation [139,140,141]. In view of the global expansion of semi-arid climates, promoting the use of this species would also help conserve important genetic resources for the continent, which are very valuable for other regions of the world [142,143]. Its wide adoption is limited largely by its low productivity. While the crop can yield in the region of 4 t/ha when grown in a research station environment, farmers working under marginal conditions typically attain only a tenth of this production level [144]. Other constraints include prolonged cooking time, susceptibility to pests and diseases and photoperiod sensitivity. In addition, as is the case for all orphan crops, seed systems are nearly non-existent and value chains need to be developed.

Bambara groundnut has been the focus of numerous studies over the last few decades. As a result, genetic resources have been characterised to varying degrees, and a range of molecular genetic tools have been developed, including DNA-based markers for some key traits [140,141,145,146,147,148,149,150]. Thus, the elements needed to develop an effective MAB programme are already in place. KT currently funds two projects focused on Bambara groundnut, the first members of the BBI: the first is a project in South Africa aiming to identify genomic regions associated with desirable agronomic traits and cooking quality as a prelude to a MAB effort to combine traits in adapted materials for southern African conditions, while the second focuses on the identification of fungal pathogens in Nigeria (https://www.kirkhousetrust.org/bambarabreedinginitiative, accessed 26 May 2024). Since this is a very new field for KT, it has been important to reach out to the existing key players and learn about ongoing research initiatives focused on Bambara groundnut to avoid duplications and capitalise on potential synergies and complementarities. To this aim, KT convened a two-day online symposium (https://www.kirkhousetrust.org/bbisymposium2024, accessed 26 May 2024), which aimed to establish collective research priorities on this crop and to discuss the most appropriate way of sharing breeding resources among partners, who committed to devote resources as a research community.

## 8. Capacity Building

Providing opportunities for scientific training is a key objective. KT has awarded 49 PhD and 58 MSc scholarships to the different research consortia. Additional training opportunities are actively sought in the form of study visits for PIs and PhD students hosted by the consultants and more experienced African PIs, independent short training throughout the year (some online), and in sessions during the yearly in-person meeting for all KT breeding PIs and PhD students. These included a course for 36 ABC project members delivered by highly qualified personnel of CIAT (the International Center for Tropical Agriculture, now known as Alliance Bioversity International—CIAT) at Kawanda (Uganda) on topics related to the use of molecular markers and plant pathology techniques. The most experienced PIs funded by KT have also delivered training sessions for their colleagues, namely a course on plant breeding in the ABC held in Kenya and a plant pathology course in the WACC hosted by INERA in Burkina Faso.

KT also funded two teaching projects that are independent of the breeding consortia: the KT Mobile Lab in Ghana, which ran from 2006 to 2022, and the molecular biology teaching lab at the University of Zimbabwe, which is still operational. The Mobile Lab was an ex-military vehicle purchased by KT in 2006 and refitted as a molecular biology laboratory (Figure 14). The Mobile Lab was shipped to Ghana and donated to the Cocoa Research Institute, Ghana (CRIG), to operate it as a training facility. Under the direction of Jemmy Takrama and, subsequently, Margaret Acheampong, the KT Mobile Lab visited research stations, universities and secondary-level institutions (to a lesser extent) across the country to provide training to scientists and students in molecular breeding and basic molecular biology techniques. Since the refitted vehicle had standing room for at most 12 persons, to enable use by a larger number of students, the equipment was moved into the main labs of the institutions visited for the training sessions to be conducted there. Over 9000 people benefited from the training opportunities it provided.

The molecular biology teaching lab at the University of Zimbabwe, under the direction of Idah Sithole-Niang, aims to increase the capacity of students and staff in the application of basic molecular biology methods applicable to many scientific disciplines. To date, close to 2800 students have received training.

## 9. Challenges and Lessons Learned

The previous sections have outlined the main successes of the KT-funded projects. There have also been many challenges. By including the building of scientific capacity in countries with limited opportunities for training as one of the key objectives, KT had to accept that progress would, at times, be slower and that there would be setbacks. Since the projects are managed remotely and some are in countries where KT’s staff and consultants cannot visit because of civil unrest, it is more difficult to provide support. KT continues to try to improve its processes for remote project monitoring and for the provision of scientific mentorship remotely. This was particularly important during the COVID-19 pandemic years when all travel stopped.

Key factors for the success of projects are the correct management of germplasm, including maintaining the purity of the recipient parents, ensuring that the target traits for introgression are not lost during the backcross process and the good management of field trials. In some projects led by PIs who were already experienced conventional breeders, a change in approach was needed for the team to use molecular markers to aid selection, as opposed to using them afterwards to verify the lines selected phenotypically.

KT invests in countries with limited economic resources, including countries experiencing political instability and comparatively high investment risk. However, since it is critical for researchers to be able to do most of the work in their home institutions, this risk has to be factored in. As part of the agreement with the host institution, the latter takes responsibility for providing suitable space and utilities, and KT provides the scientific equipment and a regular supply of consumables. Since its inception, KT has established or contributed equipment to 33 molecular laboratories in 17 countries (16 in Africa), of which at least 13 remain operational, and 5 operate independently of KT’s support. It has also funded the construction of nine screenhouses and glasshouses.

The procurement and dispatch of scientific equipment and consumables come with their own challenges, handled by a full-time staff at KT’s offices in the UK. Since most of the goods need to be imported, the challenges include navigating ever-changing customs regulations, establishing the administrative requirements of individual institutions, identifying the best suppliers either in the UK or EU or in another African country (usually South Africa), or in a combination of these; and selecting the most appropriate transport option which needs to balance cost and speed, since some of the consumables are perishable. It can take just as long and be as expensive for consumables and equipment to be delivered by African suppliers as it does when the goods are sent from the UK because African companies must also comply with import requirements. Even careful planning does not guarantee a smooth process: the delivery of a consignment of consumables can take from two to eight months (and longer in some cases). A successful consignment to a country also does not guarantee that the next one will not be problematic. Also challenging is the maintenance of equipment since not many African countries have good repair centres. Faulty or broken equipment has been brought back to the UK by KT for repair, but having previously exported the merchandise usually means that the warranty has been voided. Therefore, fixing equipment used in African countries is also much more expensive and takes longer than if the same equipment was used in the UK. Overall, the actual cost of scientific goods used in African countries is between two to four times higher than in the UK once customs and transport costs (but not repair costs) are factored in.

The difficulty in establishing advanced molecular laboratories in African countries limits the choice of relatively uncomplicated and inexpensive technologies. While molecular markers can be used very effectively to aid in the selection of traits under relatively simple genetic control, they are not as suited to aid the introgression of complex traits.

Challenges of a different nature include severe climatic conditions, with both droughts and floods compromising field experiments. The increased frequency and severity of these events render the correct management of genetic resources essential to avoid losing an entire breeding programme during extreme weather conditions.

As expected, not all the projects funded have been successful in developing and releasing improved varieties, and funding has been discontinued when it became apparent that this objective would not be met. However, this is not considered a failure since all projects generated useful data and breeding lines and contributed to the training of graduate students. With hindsight, the best approach has arguably been to initially award grants for projects with few key objectives, allowing them to naturally increase in complexity as the research team gains experience in the use of molecular biology techniques and can build on earlier achievements. It was also important for KT to engage with other organisations to understand the national and international landscape for crop improvement programmes to ensure the projects funded add value to other breeding initiatives, increasing synergies instead of duplicating efforts. Promoting the integration of the breeding projects funded in the national programmes also helps to “institutionalise” them, hoping that with time, they become less dependent on development aid and are more financed by the national, regional, or local authorities.

Ultimately, the biggest challenge faced by KT projects is making sure the improved varieties released reach the hands of farmers, the intended beneficiaries of KT’s support. Seed systems in African countries are poorly developed, suffer from poor governance, focus on formal seed systems while neglecting informal seed systems [128,137], and are mostly focused on key traded commodities such as maize. KT funded seed dissemination projects between 2016 to 2020, notably in Nigeria, Niger, Benin, Ghana, and Togo. However, these projects were discontinued since KT does not have enough resources to attain the sustainability and scale needed for impact. Therefore, a decision was made to actively seek partnerships for seed dissemination.

Increased cooperation is KT’s overall objective going forward. This includes fostering collaboration and cross-learning among projects of the same and different KT consortia. Engaging other actors in the field is also critical. Since the overall funding available for the improvement of tropical legumes is limited, a collaborative effort is needed to maximise its impact.

## Figures and Tables

**Figure 1 plants-13-01818-f001:**
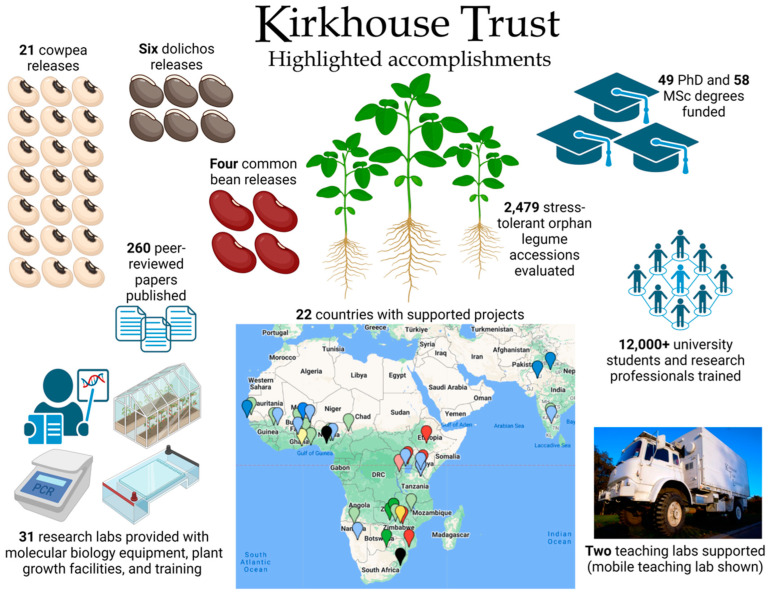
Overview of KT’s achievements 2008–2024. Legend for the map: current cowpea improvement programmes (dark green pins; Botswana, Cameroon, Zambia); past cowpea improvement programmes (light green pins; Benin, Burkina Faso, India, Ghana, Malawi, Mali, Senegal, Niger, Nigeria and Zimbabwe); current common bean improvement programmes (red pins; Ethiopia, Kenya, Mozambique, Tanzania, Uganda, and Zambia); past common bean improvement programmes (pink pins; Rwanda and Tanzania); current stress tolerant orphan legumes projects (dark blue pins; Burkina Faso, India, Namibia and Senegal); former stress tolerant orphan legumes projects (light blue pins; Ghana, India (Bangalore) Mali, Niger, Nigeria, Uganda); and teaching laboratories (black pins; Ghana (until 2023) and Zimbabwe). At the time of preparation for this manuscript, KT was reviewing potential new projects in Madagascar, Angola (common bean), and Namibia (cowpea).

**Figure 2 plants-13-01818-f002:**
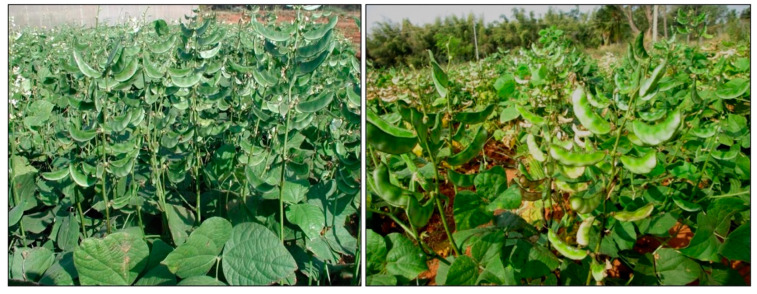
Improved dolichos lablab varieties released by the research team led by M. Byre Gowda and S. Ramesh in Bangalore, India. HA 4 (**left panel**), released in 2008, is a photoperiod insensitive, determinate and short-duration variety. It is ready for the harvest of green pods in 70 to 75 days and dry pods in 100 to 105 days, producing pod fragrance (a culinary trait valued by consumers) throughout the year. HA 4 has a potential yield of 1000–1200 kg/ha for dry seeds and 30,000–35,000 kg/ha for green pods. HA 5 (**right panel**), released in 2022, was derived from HA 4. It is a photoperiod-insensitive, indeterminate variety with pod fragrance throughout the year, and it produces a higher number of productive branches than HA 4. Its potential yield is 1100–1300 kg/ha for dry seeds and 35,000–40,000 kg/ha for green pods.

**Figure 3 plants-13-01818-f003:**
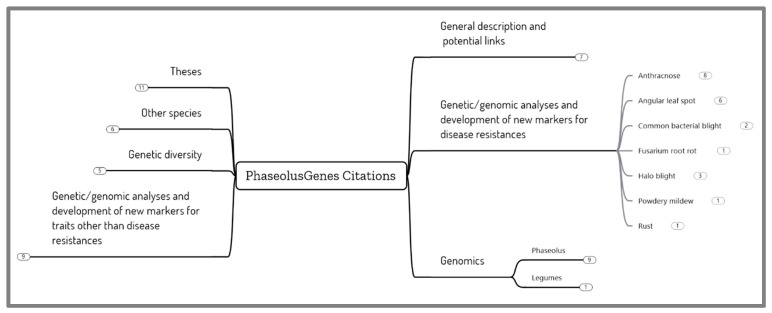
Examples of the use of the PhaseolusGenes marker database for various uses in genetics and genomics. The numbers in the figure relate to the number of published studies. General description and potential links [77,78,83,84,85,86,87]. Genetic/genomic analyses and development of new markers for disease resistances: ANT [88,89,90,91,92,93,94,95], ALS [77,96,97,98,99], CBB [100,101], Fusarium root rot [102], halo blight [103,104,105], powdery mildew [106], rust [107]. Genetic/genomic analyses and development of new markers for traits other than disease resistances [108,109,110,111,112,113,114,115,116]. Genomics: Phaseolus [80,84,117,118,119,120,121,122,123], legumes [124]. Genetic diversity [125,126,127,128,129]. Other Phaseolus species [130], non Phaseolus species [131].

**Figure 4 plants-13-01818-f004:**
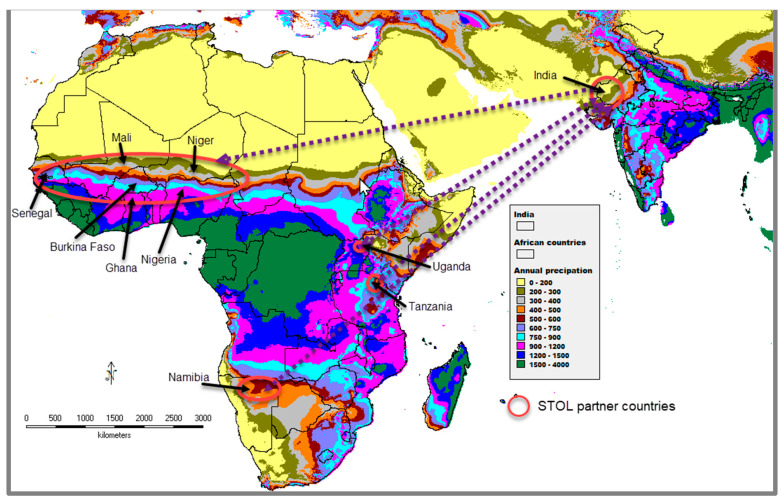
Map showing climate matching relating to average annual precipitation between India and Africa, using WorldClim Version 1.4 datasets for 2.5 min (http://www.worldclim.org, accessed on 27 May 2024). The arrows indicate the two-way exchange of germplasm among STOL partner countries.

**Figure 5 plants-13-01818-f005:**
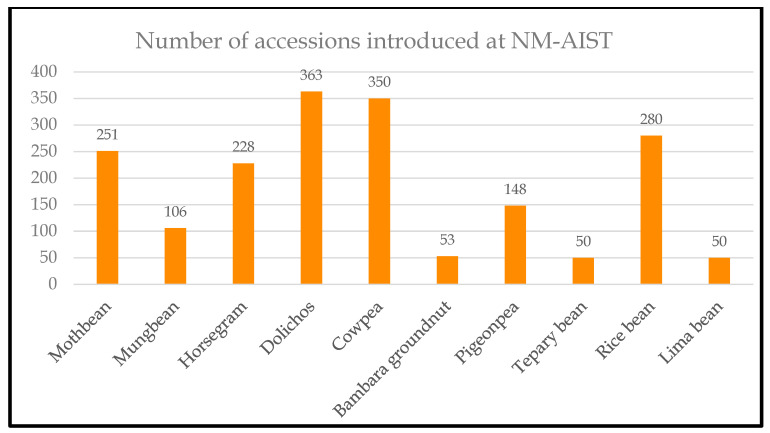
STOL germplasm collection evaluated at NM-AIST. These comprised tepary bean (50 accessions), rice bean (280), horsegram (228), moth bean (251), mung bean (106), cowpea (350), Bambara groundnut (53), pigeon pea (148), dolichos lablab (363) and lima bean (50). The number of accessions is shown on the Y axis, and the STOL crop is shown on the X axis.

**Figure 6 plants-13-01818-f006:**
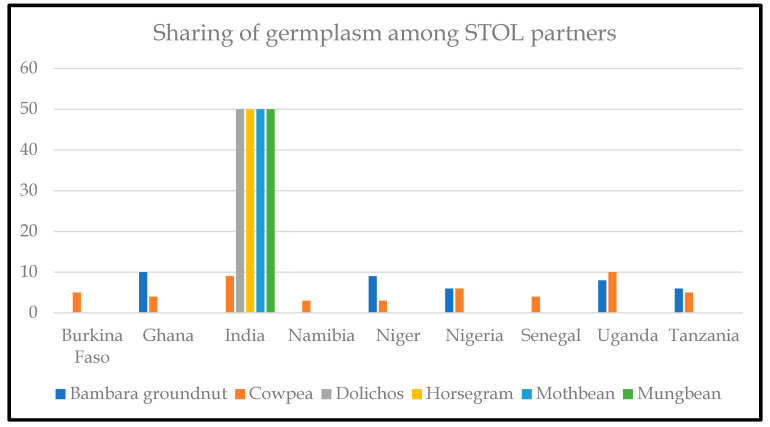
Promising varieties and germplasm are shared among the STOL partners. The number of accessions shared among countries is shown in the Y axis. The X axis shows the STOL crops contributed by each STOL country.

**Figure 7 plants-13-01818-f007:**
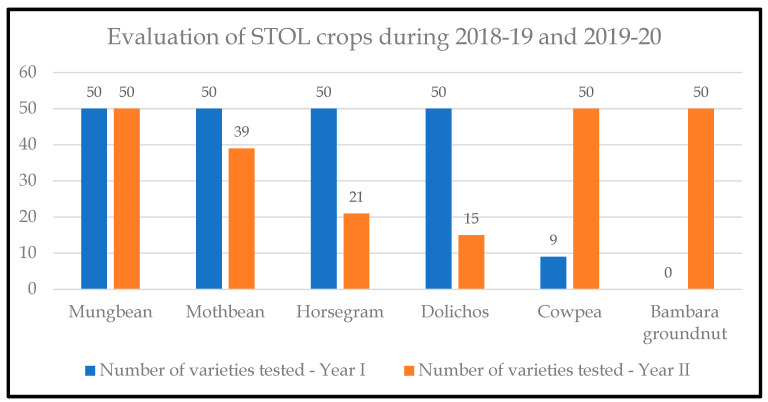
STOL crops were evaluated in the 2018–19 and 2019–2020 growing seasons. Y-axis: the number of varieties trialled; X-axis: STOL crops tested.

**Figure 8 plants-13-01818-f008:**
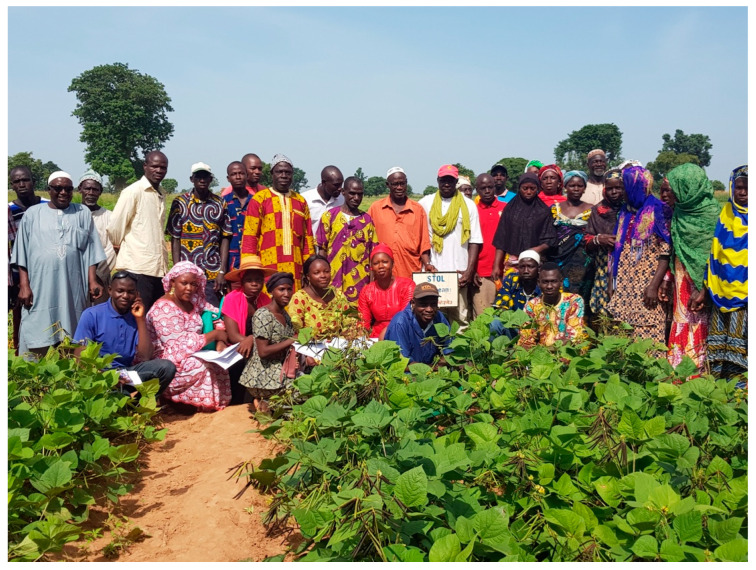
Farmer field day for STOL crops in Mali, 2019.

**Figure 9 plants-13-01818-f009:**
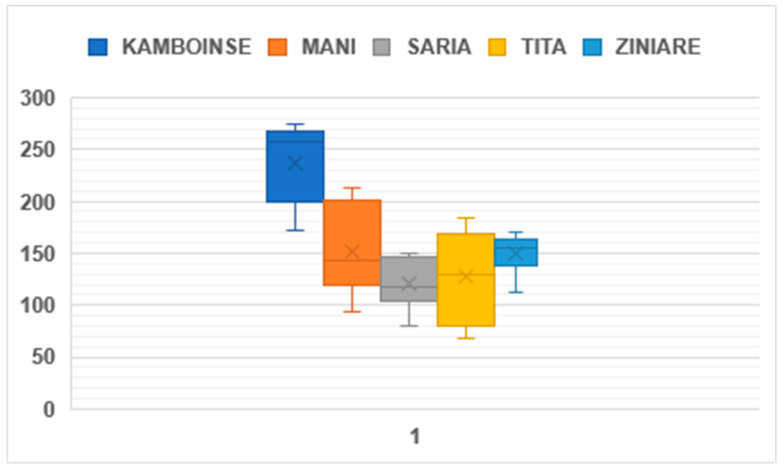
Box plot of the performance of selected mungbean varieties in the five trial locations in Burkina Faso.

**Figure 10 plants-13-01818-f010:**
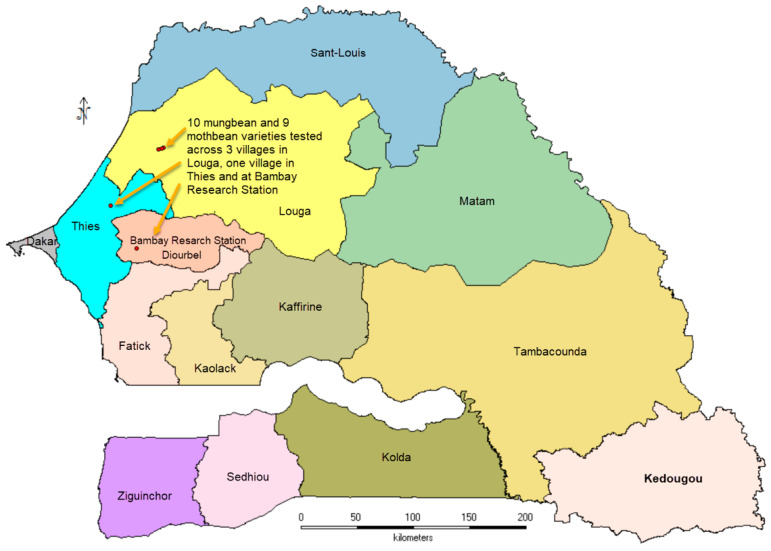
Field location for testing of mungbean and moth bean varieties in Senegal.

**Figure 11 plants-13-01818-f011:**
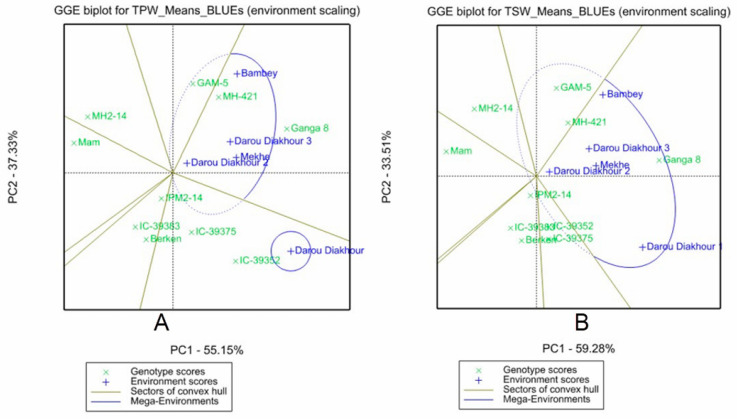
GGE biplots for (**A**) cumulative total pod weight and (**B**) cumulative total seed weight of the varieties tested in Senegal.

**Figure 12 plants-13-01818-f012:**
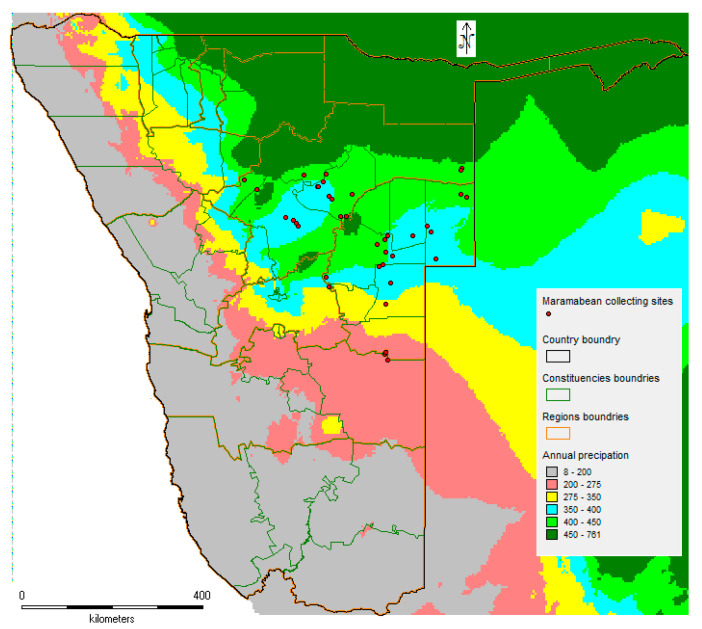
Annual rainfall pattern in the marama bean collecting sites using WorldClim Version 1.4 datasets for 2.5 min (http://www.worldclim.org, accessed on 26 May 2024).

**Figure 13 plants-13-01818-f013:**
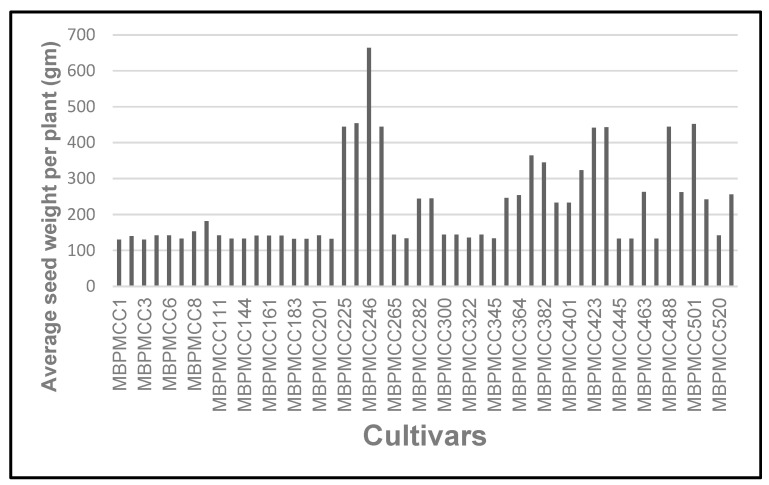
Average seed yield per plant for the marama bean cultivars tested.

**Figure 14 plants-13-01818-f014:**
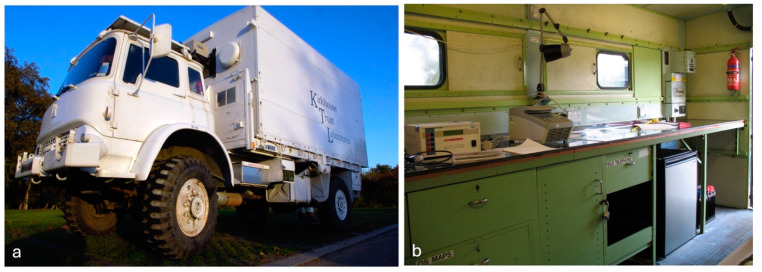
KT’s Mobile Lab was photographed upon its arrival in Ghana in 2006 (**a**); the lab benches were inside the truck (**b**).

**Table 1 plants-13-01818-t001:** Improved cowpea varieties released by the WACC and Bangalore University, India.

Yield Potential	Programme/ Developer	Outstanding Characteristics	Variety	Release Date
2000 kg/ha	Jeremy T Ouédraogo INERA, Burkina Faso	Resistant to *Striga,* large seed size, rough seed coat, white colour, 70–75 days to maturity.	Komcallé (KV442-3-25SH)	2011
2000 kg/ha	Jeremy T Ouédraogo INERA, Burkina Faso	Resistant to *Striga*, large seed size, rough seed coat, white colour, 65–70 days to maturity.	Tiligre (KVx775-33-2G)	2011
1200 kg/ha	Jeremy T Ouédraogo INERA, Burkina Faso	Resistant to *Striga*, coat, white colour, large seed size, rough seed, 60–65 days to maturity.	Nafi (KVx771-10G)	2011
1500 kg/ha	Jeremy T Ouédraogo, INERA, Burkina Faso	Resistant to *Striga*.	TZ1-Gourgou	2011
Low	M. Toure and S. Diallo IER, Mali	Tolerant to drought, resistant to *Striga*, short cooking time. Rough seed coat, white colour with brown helium, medium seed size. Dual use (forage). Early maturity.	CZ06-3-1 (Acar 1)	2015
1000 kg/ha	M. Toure and S. Diallo IER, Mali	Short cooking time, adapted to intercropping, resistant to *Striga.* Rough seed coat, white colour with brown helium, medium seed size. Medium maturity.	CZ06-2-17 (Simbo)	2015
2000 kg/ha	M. Toure and S. Diallo IER, Mali	Short cooking time, adapted to intercropping, resistant to *Striga*. Rough seed coat, white colour with brown helium, medium seed size. Early maturity.	CZ06-1-12	2015
2000 kg/ha	M. Toure and S. Diallo IER, Mali	Short cooking time, adapted to intercropping, resistant to *Striga*. Rough seed coat, white colour with brown helium, medium seed size. Medium maturity.	CZ06-4-16	2015
1500 kg/ha	M. Toure and S. Diallo IER, Mali	Short cooking time, adapted to intercropping, resistant to *Striga*. Rough seed coat, white colour with brown helium, medium seed size. Medium maturity	CZ06-1-05	2015
1900 kg/ha	Lucky Omoigui FUAM and IAR- ABU, Zaria	Resistant to Striga and *Alectra*.	FUAMPEA 1 (UAM09 1055-6)	2016
2000 kg/ha	Lucky Omoigui FUAM and IAR- ABU, Zaria	Resistant to Striga and *Alectra*.	FUAMPEA 2 (UAM09 1051-1)	2016
2500 kg/ha	Francis Kusi, SARI, Ghana	Resistant to *Striga*.	Zaayura Pali	2016
2000 kg/ha	Francis Kusi, SARI, Ghana	Resistant to *Striga*.	Soo-Sima	2016
2200 kg/ha	Francis Kusi, SARI, Ghana	Resistant to *Striga*.	Diffeele	2016
2400 kg/ha	Francis Kusi, SARI, Ghana	Resistant to *Striga* and aphid.	Wang Kae’	2016
2400 kg/ha	Francis Kusi, SARI, Ghana	Resistant to *Striga* and aphid	Kirkhouse Benga	2016
1500 kg/ha	Sobde Gonne, IRAD, Cameroon	Resistant to *Striga*.	Lori-2 (IR15-MA-02)	2019
1500–2000 kg/ha	Sobda Gonne, IRAD, Cameroon	Resistant to *Striga*.	Lori-3 (IR15-MA33)	2019
1671 kg/ha	Lucky O. Omoigui, Joseph SarwuanTarka, University Makurdi, Nigeria	Medium maturity group, high-yielding, resistant to *Striga*, large brown seeded.	FUAMPEA 3 (UAM14-122-17-7)	2023
1510 kg/ha	Lucky O. Omoigui, Joseph SarwuanTarka, University Makurdi, Nigeria	Medium maturity group, high-yielding, resistant to *Striga*, large brown seeded.	FUAMPEA 4 (UAM14-123-18-3)	2023
1300–1400 kg/ha	H.C. Lohithaswa, University of Agricultural Sciences, Bangalore, India	Resistant to bacterial leaf blight, cowpea mosaic virus, and dry root rot.	KBC-12	2024

**Table 2 plants-13-01818-t002:** Improved common bean varieties released by the ABC.

Yield Potential	Programme/ Developer	Outstanding Characteristics	Variety	Release Date
1700–3100 kg/ha	Yayis Rezene, SARI	Small dark red; Tolerant to ALS, CBB and BSM	Key-Wolaita	2023
1800–2400 kg/ha	Yayis Rezene, SARI	Large red mottled; Tolerant to ALS, CBB and BSM	Key-Burre	2023
2200–3800 kg/ha	Stanley Nkalubo and Annet Namayanja	Large cream with purple speckles; Resistant to root rot; Tolerant to ALS, CBB and ANTH, Susceptible to BSM and Rust.	NABE 12CR	2024
1500–2700 kg/ha	Stanley Nkalubo and Annet Namayanja	Medium red; Resistant to ANTH; Tolerant to ALS, CBB and Root rot, Susceptible to BSM and Rust.	NABE 14R	2024

**Table 3 plants-13-01818-t003:** Mean and variance variability for various morphological traits in the marama bean germplasm tested.

Component	Days to Flowering	Days to Maturity	Number of Seeds Per Pod	Yield (g)	1000 Seed Weight (g)
Mean	32.56	90.28	1.62	226.8	3060
Standard Error	0.242705835	0.54213	0.080255	17.91075	25.41412
Median	34	90	2	144	3020
Mode	34	90	2	144	3000
Standard Deviation	1.716189419	3.83347	0.567486	126.6482	179.705
Sample Variance	2.945306122	14.6955	0.322041	16,039.76	32,293.88
Kurtosis	−1.396759409	2.60796	−0.76405	1.59793	0.179794
Skewness	−0.585593235	−1.0045	0.201565	1.461254	0.83115
Range	4	16	2	530	600
Minimum	30	80	1	130	2800
Maximum	34	96	3	660	3400
Sum	1628	4514	81	11,340	153,000
Count	50	50	50	50	50
Confidence Level (95.0%)	0.487735636	1.08946	0.161278	35.99301	51.07159

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
