# Peer review of "The Kirkhouse Trust: Successes and Challenges in Twenty Years of Supporting Independent, Contemporary Grain Legume Breeding Projects in India and African Countries"

_plants, 2024, doi:10.3390/plants13131818_

Round 1
Reviewer 1 Report
Comments and Suggestions for Authors
This is a clear and logically presented review of KT's extremely timely and pressing work with legume breeding and agri partners across Africa, and to a lesser extent Asia, over the past twenty years. It is very useful summary of KT's past successes and its learnings, as well as the directions it is moving in. I have only a few very minor comments and suggestions that could hopefully help the reader.
P2: In the first line/sentence of p2, it could be useful to mention the Africa Biogenome project which shares similar aims wrt molecular capacity building.
P4: Figure 2 legend: 'pod fragrance'. It could be useful to explain this. Is it a desired trait for growers/consumers? Also typo: 'in determinate'.
P5: I think historical cowpea 'migrations' are now better understood. Could be worth citing , , . Genetic, textual, and archeological evidence of the historical global spread of cowpea (Vigna unguiculata [L.] Walp.). Legume Science. 2020; 2:e57. https://doi.org/10.1002/leg3.57
P9: as well as reducing fuel use, a key consideration wrt cooking time is the reduction in water use.
P10: last paragraph. typo: 'for o identification'
P11: 'objectives of the ABC: anthracnose'
P15: Why was the 2018 MoU not including Senegal when it seems to be a key partner? Or is this just an omission? Why Kenya instead?
P15: 'proposals for release'
P16-17: Y axis labels missing
P18: ''all varieties' performances''
P19: Both axis labels missing from this graph.
P20: Fig 11 very small font sizes
P20: 'Marama bean (Tylosema esculentum)' . Here and elsewhere in the manuscript, it could be useful to include the binomial when a species is first mentioned (or mentioned again after a while).
P20: and elsewhere, figure legends run over to the other pages.
P21: Fig 13 Y axis label missing, and X axis label is incorrect and he instead meant to be in Y, X should be 'cultivar'.
P22: 'semi-arid climates'
P23: 'which ran from 2006...' and delete 'subsequently' which appears twice.
P24: Why was the mobile lab discontinued? And is there a desire to reinstate it?
P5 'focuses'...and 'Current member not on the author line' or something along those lines?
Comments on the Quality of English Language
High quality English throughout
Author Response
Responses to Reviewer 1
We thank the reviewer for the careful editing and thoughtful comments and suggestions for improving this manuscript.
P2: In the first line/sentence of p2, it could be useful to mention the Africa Biogenome project which shares similar aims wrt molecular capacity building.
- We thank the reviewer for this suggestion and agree that there are similarities in the objectives of this project and KT’s. We have not included a reference in the text since the reports of the Africa Biogenome pilot projects are still not available in their website.
P4: Figure 2 legend: 'pod fragrance'. It could be useful to explain this. Is it a desired trait for growers/consumers? Also typo: 'in determinate'.
- This has been explained in more detail in the figure legend (“a culinary trait valued by consumers”). The typo has been corrected, thanks.
P5: I think historical cowpea 'migrations' are now better understood. Could be worth citing Herniter IA, Muñoz-Amatriaín M, Close TJ. Genetic, textual, and archeological evidence of the historical global spread of cowpea (Vigna unguiculata [L.] Walp.). Legume Science. 2020; 2:e57. https://doi.org/10.1002/leg3.57
- This reference has been included.
P9: as well as reducing fuel use, a key consideration wrt cooking time is the reduction in water use.
- Thank you for the suggestion, this has been included.
P10: last paragraph. typo: 'for o identification'
- This has been corrected
P11: 'objectives of the ABC: anthracnose'
- This has been corrected.
P15: Why was the 2018 MoU not including Senegal when it seems to be a key partner? Or is this just an omission? Why Kenya instead?
Senegal joined the STOL programme in 2020. This has been included in the text.
The initial STOL partners had already been collaborating with KT/ Prem Mathur (the scientific leader of STOL), on other projects before the start of this programme, so it was easier to establish the agreements. Kenya has only a small area suited to the cultivation of STOL crops, so trials were discontinued after the initial year.
P15: 'proposals for release'
- Typo corrected, thanks.
P16-17: Y axis labels missing
- Indicated in the legend
P18: ''all varieties' performances''
- Corrected, thank you.
P19: Both axis labels missing from this graph.
- Indicated in the legend.
P20: Fig 11 very small font sizes
-The figure has been edited with larger font sizes.
P20: 'Marama bean (Tylosema esculentum)' . Here and elsewhere in the manuscript, it could be useful to include the binomial when a species is first mentioned (or mentioned again after a while).
- All binomial of all the species are included when they are first mentioned in the text.
P20: and elsewhere, figure legends run over to the other pages.
- We believe this is something that will be rectified by the journal during the final stages of setting the layout?
P21: Fig 13 Y axis label missing, and X axis label is incorrect and he instead meant to be in Y, X should be 'cultivar'.
- This has been corrected, thank you.
P22: 'semi-arid climates'
- This has been corrected.
P23: 'which ran from 2006...' and delete 'subsequently' which appears twice.
- This has been corrected.
P24: Why was the mobile lab discontinued? And is there a desire to reinstate it?
- The Mobile Lab was discontinued for a number of reasons. The vehicle itself had reached the end of its life. The number of students served by the Mobile Lab was high, not all students could engage in hands on activities. Since the start of the programme, high quality online learning modules became available, so it was felt that this initiative was no longer adding significant value. There is no intention to reinstate it, although providing training opportunities remains one of KT’s key objectives.
P5 'focuses'...and 'Current member not on the author line' or something along those lines?
- The typo has been corrected. The sentence changed to “Current KT staff members”. These roles are administrative.
Reviewer 2 Report
Comments and Suggestions for Authors
Comments: It is no doubt that there are great achievements on these KT funded projects. It will certainly provide many benefits to India and those African countries which involve in these projects development and applications.
Suggestions : KT continues to support these projects further to more countries globally , it would be more beneficial to the world sustainable agriculture since the legumes crop play very important roles on the sustainable environment.
Comments on the Quality of English LanguageThere are some informal writing need to re-edit.
probably better add one more final summary paragraph.
Author Response
We thank the reviewer for the suggestions and comments.
While we agree on the global importance of legumes, KT does not have the resources needed to invest more widely.
We respectfully disagree with the statement that the language used is informal.
We have considered adding an additional final summary paragraph but decided against this as we do not feel this would improve the manuscript.
Reviewer 3 Report
Comments and Suggestions for Authors
The manuscript is a comprehensive review of projects funded by the Kirkhouse Trust (KT) in two decades for supporting independent, contemporary grain legume breeding projects in India and African countries. The authors have introduced KT’s funding model, outlined the main successes of the KT-funded projects, and points out the challenges confronted by the project and the lessons learned.
I didn’t find major flaws of this review but have a few suggestions.
1) Table legends should be placed above the table.
2) Whether the PhaseolusGenes marker database mentioned in sub chapter 5.2 is public? If there are websites please provide. The data of the markers described in this sub chapter are better to be presented as tables.
Author Response
We thank the reviewer for his comments and suggestions.
1) Table legends should be placed above the table.
- We do not have any objection to the placement of the legends of the tables and would like to defer to the journal's editorial policy.
2) Whether the PhaseolusGenes marker database mentioned in sub chapter 5.2 is public? If there are websites please provide. The data of the markers described in this sub chapter are better to be presented as tables.
- As noted in the manuscript, the PhaseolusGenes database has been incorporated in the PhaseolusMine in the Legume Information System (LIS). A hyperlink to the website has been included.